# The Impact of Anosmia on Intimacy: A Linear Mixed-Effects Model Analysis of Sexual Wellbeing Following COVID-19

**DOI:** 10.3390/ijerph22101597

**Published:** 2025-10-21

**Authors:** Arianna Miclet, Daniele Mollaioli, Marco Lauriola, Giacomo Ciocca, Andrea Sansone, Emmanuele A. Jannini, Erika Limoncin

**Affiliations:** 1Department of Dynamic and Clinical Psychology, and Health Studies, Sapienza University of Rome, 00185 Rome, Italy; 2Department of Clinical and Experimental Medicine, University of Messina, 98125 Messina, Italy; 3Department of Psychology, Sapienza University of Rome, 00185 Rome, Italy; 4Department of Systems Medicine, University Tor Vergata, 00133 Rome, Italy

**Keywords:** COVID-19, anosmia, sexual function, orgasm

## Abstract

Olfaction is thought to have a role in intimate relationships and sexuality, despite the major roles that other senses, like sight, might have for human beings. Moreover, one of the specific symptoms of the COVID-19 infection, a disease largely impacting human sexuality at various levels, has been the absence of smell. This study aimed to examine the impact of olfaction on partner selection and on different phases of sexual response. Moreover, we evaluated whether the absence of smell could influence aspects of sexual function, such as orgasmic intensity, by comparing subjects with and without COVID-19 olfactory dysfunction. A group of 259 participants has been initially enrolled through social media. Of the 138 sexually active subjects included in the final analysis, 61 (44.2%) reported olfactory dysfunction (anosmia group) and 77 (55.8%) did not (no-anosmia group). Perceived orgasmic intensity was evaluated retrospectively at two time points—during the COVID-19 infection and in the four weeks preceding the evaluation—using the male and female versions of the Orgasmometer. The results revealed a significant interaction between time and olfactory condition, indicating opposite recovery trajectories. Participants without anosmia showed a significant improvement in orgasmic intensity post-COVID-19 infection; in contrast, the anosmia group reported a significant decline over the same period. Although the presence of anosmia was not a direct predictor of orgasmic intensity *per se*, it critically determined the orgasmic intensity evolution over time. These findings suggest that, beyond its perceived importance, the actual absence of olfaction may have a specific and negative long-term impact on core components of the sexual experience, such as the intensity of orgasm.

## 1. Introduction

Human beings rely on perceptions and cognitions to interact with the world. Among perceptions, the sense of smell is the sensory system that enables the detection and discrimination of volatile molecules in the air. These molecules activate specific receptors located in the nasal mucosa, and the information is transmitted to the olfactory bulb and subsequently integrated into limbic structures, which connect to memory and emotion processes [1,2]. However, these chemicals have diverse structures, and those structures are somehow perceived as having different odours [3]. Some recent theories suggest that smell is not an intrinsic property of molecules, but rather a perceptual construction of the brain [4].

From a neurophysiological level, it is known that the amygdala and hippocampus play a key role in linking odours to emotions and memories, contributing to the integration between olfactory perception and affective states [5,6]. The sense of smell is crucial in several aspects of human life, including parental recognition, food preferences, and survival [7]. Its loss, known as anosmia, is associated with increased environmental risk, such as the inability to detect spoiled food or hazardous substances [8].

Recent studies have shown a link between the sense of smell and sexuality. Olfactory signals operate on an unconscious level and can elicit emotional and physiological responses involving all phases of the sexual response cycle [9]. In women, olfactory cues can enhance sexual desire. Studies demonstrate that scents related to sexual arousal can heighten emotional receptivity and physiological readiness, suggesting that pleasant odours may facilitate the “Desire” and “Excitement” stages in Basson’s model [10,11]. These stages, together with Plateau, Orgasm, and Resolution (D.E.P.O.R.), represent the components of the so-called “linear and circular model” explaining the sexual response phases. Higher olfactory sensitivity is positively correlated with a more fulfilling sexual experience, indicating that scent can enhance intimacy and connection during sexual interactions [12]. In males, olfactory input is found to correlate with sexual arousal and sexual function. Men with better olfactory function report higher levels of sexual pleasure, suggesting that olfactory stimuli also enhance their sexual responsiveness [11,13]. Furthermore, even if without a strong scientific evidence, scents in humans may operate as pheromonal signals subtly communicating partner compatibility, thus impacting tactile and experiential dimensions of the sexual response, particularly within the “Excitement” and “Orgasm” phases [14].

On the other hand, people suffering from acquired olfactory loss are affected in sexual experience (i.e., missing their partner’s body odour) as well as in sexual desire, even if the decreased sexual desire was found to be mediated by depression levels [15]. In another study, it was found that participants with high olfactory sensitivity experienced higher pleasantness of sexual activities, and women with high olfactory sensitivity had a higher frequency of orgasms during sexual intercourse [12]. Men without the sense of smell were reported to adopt to a lesser extent explorative sexual behaviours and to have fewer sexual partners, while women without a sense of smell reported a reduced partnership security [16]. Smelling the partner seems to be important for women. Not being able to do this is related to reduced security on the attachment bond. A recent review found evidence that olfactory dysfunction is linked to decreased interest, enjoyment, and participation in sexual activities, as well as impacting the individual’s perception of their potential sexual partners and their own interest in sex [17]. At the same time, olfaction contributes to the quality and quantity of sexual behaviour [12] and to specific phases of the sexual response, like that of sexual arousal [18], which are mediated by body odour perception.

Another sexual aspect related to the sense of smell is orgasm. In fact, olfactory signals can trigger reflexive physiological responses that enhance sexual arousal and orgasmic potential [19,20]. The presence of certain scents can elicit conditioned responses, where individuals learn to associate specific odours with erotic or intimate contexts, leading to heightened arousal in subsequent encounters. This conditioning emphasizes how olfactory cues can reinforce sexual desire and contribute to the orgasmic experience [21]. The feedback loop created between these sensory stimuli and emotional states cultivates an acute awareness of one’s physical and emotional readiness during sexual moments. Smell disorders have been observed in people affected by COVID-19. The prevalence of olfactory dysfunction was 55.44, 56.05, 50.42, 28.83, and 26.50% in Europe, the Americas, the Middle East, Asia, and Africa, respectively [22]. The prevalence of anosmia in a recent meta-analysis from a pool of 107 studies was rated at an average of 38.2% [23]. A case study of a 29-year-old Italian woman reported a distressing COVID-19-related loss of olfaction and a taste dysfunction, which were severely impacting her personal and sexual life [24]. Without the ability to perceive her partner’s scent, a prominent component of her sexual attraction to him, and of her sex drive, was lost.

In 2022, Sansone et al. proposed a Sexual Long COVID disease (SLC), referring to a variety of symptoms affecting different organs reported by people following recovery from a SARS-CoV-2 infection, where neuropsychiatric, respiratory, cardiovascular, and endocrine complications of the COVID-19 disease may constitute risk factors for sexual dysfunctions and erectile dysfunction (ED). Among the neuropsychiatric complications, anosmia and ageusia might contribute to the worsening of sexual function in such patients [25].

Scientific data about the association of olfaction and sexual function seem to indicate that this sense, probably, can orient the partner’s selection and sexual behaviour.

Although the acute phase of the pandemic has subsided, the widespread incidence of olfactory dysfunction post-infection has created a unique natural milieu, offering an unprecedented opportunity to investigate the consequences of smell loss on a large scale. Therefore, by studying this specific population, our findings can be generalized, providing broader implications for understanding the impact of anosmia from any aetiology (e.g., post-viral, traumatic, or idiopathic) on sexual wellbeing and intimacy. Thus, COVID-19 gave the possibility to address if and how olfactory dysfunctions may impact specific aspects of sexual function, such as orgasmic experience. For all these reasons, this study had two aims:To quantitatively examine the association of olfaction on partner’s selection and on different phases of the sexual response according to D.E.P.O.R. (Desire, Excitation, Plateau, Orgasm, Resolution) model [18].To evaluate if the absence of smell (i.e., anosmia) had a role on the orgasmic experience by comparing subjects with COVID-19 anosmic symptoms and subjects without anosmia.

## 2. Materials and Methods

### 2.1. Participants and Procedure

The data for this paper come from a survey of a sample of the Italian population aged between 19 to 65 years old. The sample has been collected through online platforms, through the main social networks (Facebook and Instagram) and websites (Survey sharing, Surveycircle). A total of 259 participants were recruited, 167 females and 92 males. Inclusion criteria comprised: an age > 18, COVID-19 infection, the absence of self-referred psychotic diseases and being sexually active both during the COVID-19 infection and in the 4 weeks prior the data collection. According to inclusion criteria, a final sample of 138 subjects (males: 42/138; 30.4%; females: 96/138; 69.5%) were analysed. A demographic information questionnaire and an informed consent form (ICF) were developed by the principal investigator. All the participants were required to fill up the ICF before taking part in the study. This study has been approved by the Ethical Committee of the Department of Dynamic and Clinical Psychology, and Health Studies, of Sapienza University of Rome.

### 2.2. Measures

Several demographics, COVID-19-related questions, and the auto-referred mental and physical well-being were assessed through self-report questions. Data on general health status, physical well-being (smoking habits, Body Mass Index (BMI), medical conditions, genetic conditions bearing on the olfaction-Kallmann syndrome), and any chronic illnesses or disabilities of the subjects were entirely based on the participants’ self-assessment. Among COVID-19-related questions, the presence of self-referred olfactory symptoms (incomplete or absent ability to detect and fully recognize the odour of the partner) was asked. Moreover, the duration of olfactory symptoms was evaluated through the following options: “having had olfactory symptoms only during the illness”, “having had olfactory symptoms up until 1 month after the illness”, “having had olfactory symptom up until 2 months after the illness”, “having had olfactory symptoms up until 6 months after the illness and still having olfactory symptoms”. Mental well-being questions assessed the presence of self-reported as well as diagnosed mental disorders (depression, anxiety). Physical well-being questions inquired about the smoking habits (if the subject is currently smoking or has ever smoked and how long ago, they stopped) and the eventual presence of non-communicable diseases (chronic diseases that are not passed from person to person), as well as medical conditions. Individuals were finally asked to rate the importance of olfaction and different types of smell for the selection of a partner and its role on sexuality, with both qualitative questions as “Do you think that the smell plays a (positive) role in partner’s selection/sexual response?”, “In which phase of the sexual response?”, “What type of odour has a role in the partner’s selection/sexual response?” and quantitatively rate the importance on a scale from 0 to 10, later arbitrarily summarized in macro-categories as “low” (0–3), “medium” (4–7), or “high” (8–10) importance. The orgasmic experience, intended as perceived orgasmic intensity during sexual intercourse, was evaluated with the male and female versions of the Orgasmometer [26,27]. The Orgasmometer is a single-item psychometric tool, derived from the Visual Analogue Scale [28], useful to record and give a quantitative measure of subjective perception of orgasm intensity on a Likert scale, with points ranging from 0 to 10. Both validation studies reported high reliability (0.96 for males and 0.93 for females) and the ability to discriminate subjects with and without sexual dysfunctions [26,27].

### 2.3. Statistical Analysis

All statistical analyses were performed using the R software, version 4.5.0 [29]. Continuous variables were normally distributed and were expressed as means and standard deviations. Dichotomous variables were condensed as absolute, and relative frequencies and differences across groups were determined by χ^2^ test with Bonferroni correction for multiple comparisons. Missing data were handled using the multiple imputation procedure via the “mice” free software R package (v4.1.2; R Core Team 2021) [30]. Five imputed datasets were generated (m = 5) over 50 iterations (maxit = 50), employing the Predictive Mean Matching (PMM) method to impute the continuous variables. To investigate the impact of anosmia on the Orgasmometer score over time (during and after COVID-19 infection), a linear mixed-effects model (LMM) was applied with “lme4” R package [31]. The model included the Orgasmometer score as the dependent variable, with time, the presence of anosmia, gender, age, self-referred anxiety and depression as fixed effects. A Time*anosmia interaction term was also included to assess whether the change in score over time differed between the two groups (anosmia/non-anosmia). A random intercept for each participant was added to account for the repeated measures. The results from the imputed models were then pooled to obtain final, corrected estimates and standard errors.

## 3. Results

The study sample was divided according to self-referred presence of anosmia.

Concerning the importance of body smell for sexuality, the responses of all enrolled participants were considered. However, no responses were obtained to some of the questions. Most sample participants reported the importance of smell in the partner’s selection and during the sexual response. Overall, 91% of the total sample (233/256) reported that smell is important in partner’s selection (*p* < 0.001), and the 90.7% of the subjects (233/257) reported that the sense of smell has a role during the sexual response.

Of the total sample of this study, 88.9% of males (80/90) and 92.2% of females (153/166) who responded to the question declared that smell has an important role in the selection of a sexual partner (*p* = 0.51). It was also investigated what type of smell influenced the partner’s selection. For all the samples, the partner’s skin (75.55% of males; 81.53% of females; *p* = 0.2) was the most important, followed by the product used for body care (57.77% of males; 68.86% of females; *p* = 0.06). The genital smell has a role in partner’s selection for 35.2% of males and 31.7% of females (*p* = 0.5). Very few people said that no type of smell is thought to influence a partner’s selection (below 8%). A social desirability bias might influence the frequency of responses concerning the bad smell, which, still, for males, has higher rates than the “no-smell” category. In all the samples, the above types of smell were of medium to high importance.

The phase of sexual response in which the perceived importance of smell is reported to be greatest is desire for women (90.9%), followed by arousal (72.72%) and resolution (34.09%), while for men the phase in which smell is most important is instead arousal (77.5%), followed by desire (75%) and resolution (30%). For both women and men, orgasm appears to be subjectively little influenced by smell. The type of smell that had the most influence on excitement for both males and females was the skin odour (75.28% of males; 82.5% of females; *p* = 0.16), followed by the odour of the genitals for men (60.67%) and the products used for women (51.20%). More men than women found the odour of the genitals important for the sexual response (*p* = 0.037). No gender difference was found for the products used (*p* = 0.92). The importance given by most participants was medium-high both in partner selection and during sexual response.

Table 1 lists the demographic characteristics of the subjects considered for the final quantitative analysis (*N* = 138). All these subjects satisfied the inclusion criterion of being sexually active during the COVID-19 pandemic.

A linear mixed-effects model was performed to examine the effect of olfactory symptoms on the Orgasmometer score over time, while controlling for potential covariates. The model revealed a significant main effect of time, with a positive coefficient (β = 1.04, *p* = 0.011) (Table 2), indicating a significant increase in the Orgasmometer score for the subjects without anosmia from T0 (during COVID-19 infection) to T1 (after COVID-19 infection). This change, however, was qualified by a highly significant interaction between “time” and “anosmia” (β = −2.94, *p* < 0.001).

As visually represented in Figure 1, the anosmia group showed a marked decrease in their Orgasmometer score over the same period, suggesting that the positive temporal change was absent for this group. No significant main effects at T0 were observed for the presence of anosmia (β = −0.72, *p* = 0.183), gender (β = 0.56, *p* = 0.135), age (β < 0.001, *p* = 0.984), self-referred anxiety (β = −0.17, *p* = 0.628), or depression (β = 0.46, *p* = 0.185), indicating that these variables did not independently account for a significant portion of the variance in the Orgasmometer score within the model.

## 4. Discussion

To our knowledge, the total prevalence of smell disturbances alone in the Italian population has not been adequately investigated, though studies on COVID-19 patients have reported rates of smell and taste alterations ranging from 34% [32] to 64.4% [33]. While not representative of the general population, likely due to a selection bias favouring participants with a history of COVID-19, the present study aimed to leverage this clinical context.

The primary aim of this study was to evaluate the importance of olfaction in selecting a sexual partner and its role in specific phases of sexual response. Our results show that for most of the sample, olfaction has an important role in both the partner’s selection and in several phases of sexual response. Interestingly, specific odours seem to play a role in sexuality, such as a partner’s body odour, products used for body care, and, to a lesser extent, genital odour. These qualitative data confirm what literature has previously found [34,35,36].

The secondary aim of this study was to determine whether COVID-19-induced anosmia had an impact on a specific component of sexual function, namely the perceived intensity of orgasm. Hence, we compared subjects with and without olfactory dysfunction at two time points: during the infection and after recovery.

The central finding of our study is the significant interaction between time and the presence of anosmia on orgasmic intensity. Our results indicate that the two groups followed dramatically divergent trajectories post-infection. While individuals without olfactory dysfunction reported a significant recovery in orgasmic intensity—a finding consistent with a general return to wellbeing—participants with persistent anosmia showed a significant and paradoxical decline. This result aligns with the literature suggesting that olfactory loss negatively affects sexual experience [12,13], but critically extends this impact to the specific domain of orgasm. Importantly, this effect persisted even when controlling for self-reported anxiety and depression, suggesting that the decline is more directly linked to the sensory deficit itself rather than being a secondary consequence of poorer mental health. These data are in line with literature evidence, which has demonstrated the lack of association between the improvement in psycho-physical or self-rated olfactory function and the reduction of depressive or anxiety levels [35].

The link between olfaction and sexual function has been frequently studied and evidenced in the literature [12]. Concerning the reduction or lack of olfaction, it has been evidenced that patients with olfactory dysfunctions may report higher levels of insecurity in romantic relationships [15]. Due to this negative impact, it is easy to hypothesize how anosmia can worsen sexual health. Interestingly, improvements in olfactory function did not predict changes in sexual frequency or desire. However, improved psychophysical olfactory function is associated with more favorable sexuality outcomes and may positively predict sexual frequency [35].

Other studies have demonstrated that reduced olfactory ability was associated with lower sexual motivation, reduced sexual satisfaction, and low orgasmic frequency [12,37]. In our case, on the contrary, orgasmic intensity was higher during the COVID-19 period, rather than during its recovery.

The decline in the anosmia group evidenced in our sample raises questions about the limits of sensory compensation. While it is often theorized that other senses like touch or sight might compensate for olfactory loss [38], our data suggest that for a deeply integrating and multi-sensory experience like orgasm, this compensation may be insufficient. The absence of a partner’s scent—a primal and intimate cue—may create a subtle but significant disconnection that other sensory inputs cannot fully overcome, thus diminishing the peak sexual experience.

If, on one hand, the sexual improvement evidenced in the non-anosmia sample is in line with the theory of the recovery of general well-being after the disease, the significant reduction in the orgasmic intensity observed in the anosmia group does not follow this theory. During infection, symptoms such as fatigue, stress, and anxiety may reduce sexual desire and satisfaction. The end of the illness period could therefore contribute to a natural recovery of sexual function. In the case of the anosmia group, on the contrary, different symptoms may have positively impacted the orgasmic intensity during the COVID-19 pandemic, and negatively impacted after the symptoms have remitted. This data could be counterintuitive if we do not read the sexual function in a multifactorial perspective. One possible interpretation concerns sensory compensation and attention: during anosmia, people may have unconsciously compensated for losing the sense of smell by intensifying their attention on other aspects of sexuality, such as touch, imagination, and body sensations. In neuroscience, it is known that when one sense is diminished, others can become more sensitive by adaptation [17,35]. We also know that anosmic individuals can better integrate auditory and visual information than control participants [36].

Following a different perspective, Sansone et al. demonstrated the existence of the so-called “Sexual Long COVID” disease (SLC). This condition, which can manifest itself after COVID-19 remission, is characterized by multiple symptoms, such as brain fog, fatigue, sensory, respiratory, and cardiovascular dysfunctions, and endocrine alterations. All these symptoms are associated with impaired sexual function through multiple pathogenic mechanisms [25,39]. Hence, the reduction in the orgasmic intensity for the anosmia group may be interpreted as a sign of the prolonged symptomatology that bears on the sexual function.

We could finally assume, in an easy way of thinking, that anosmia, which in some cases was present also after a long period after the COVID-19 remission, had determined the decrease in the orgasmic intensity. However, we know that the orgasm is a complex, multifactorial experience, which cannot be explained by a singular variable [40]. Following a bio-psycho-social model, we would hypothesize that other psychological or relational variables, such as the proximity with the partner during the lockdown, or the shared experience of the coping with a traumatic event such as the COVID-19 pandemic [41], may have improved the quality of the intimacy in terms of a better sexual pleasure, resulting so in this only apparently paradoxical result.

## 5. Conclusions

The results of this study suggest that although olfaction is perceived as important for intimate relationships, its loss due to COVID-19 infection significantly impacted the orgasmic intensity only during the recovery period. The discrepancy between the infection and recovery period could be due to the ability of other senses to compensate for olfactory loss during the infection, or due to the influence of other psycho-relational variables, which may have contributed to ameliorating the relational quality during the lockdown. Even without specific answers, the role of olfaction in the perception of intimacy and connection with a partner remains an area of great interest, which deserves further investigation in future research. It is important to emphasise that assessing the impact of olfaction on sexuality is not equivalent to assessing its absence. While the presence of olfaction may enrich the sensory and relational experience, its absence does not necessarily imply an impairment of sexual function, as other mechanisms may intervene to compensate for this loss.

### Limits and Future Research

This study presents several limitations that should be considered when interpreting the findings. A primary issue is the reliance on self-reported data, which may introduce biases such as social desirability bias—where participants may overestimate the role of smell in relationships due to societal expectations and study related expectations—and confirmation bias, as individuals with strong beliefs about the importance of olfaction might be more likely to participate or answer in a way that aligns with their preconceived notions. Moreover, retrospective self-reports can be affected by memory recall errors, further complicating the reliability of the responses.

Another limitation concerns the absence of objective olfactory assessments. The classification of participants as having or not having olfactory dysfunction was based on subjective reports, which may not accurately reflect their actual olfactory capabilities. Research has shown that objective olfactory tests, such as the Sniffin’ Sticks, tend to detect a higher prevalence of olfactory dysfunction than self-reports. To improve diagnostic accuracy, future studies should incorporate standardized smell tests. Additionally, the sample size was relatively small, which limits the statistical power needed to detect subtle effects. The recruitment method, which relied on online platforms, may have introduced selection bias, as individuals with a particular interest in the topic may have been more likely to participate, reducing the generalizability of the findings.

Beyond methodological constraints, this study did not thoroughly explore the interplay between anosmia and relational or psychological factors that may influence sexuality. Civil status as a variable (e.g., single, married) was insufficient to capture the nuances of different relationship dynamics, such as long-term partnerships, casual dating, and polyamorous relationships. Investigating how relationship type and dynamics influence the role of olfaction in intimacy would provide deeper insight into this complex association.

Future research should integrate objective smell assessments to enhance the reliability of olfactory classification. In addition, longitudinal studies tracking changes in sexual function over time could determine whether the effects of anosmia on sexuality are transient or persistent. Beyond evaluating the direct effects of olfaction on sexual response, future studies should explore its role in partner recognition and bonding, as many participants in this study reported that smell was important in choosing a partner, despite the lack of a significant impact on sexual function. This data suggests that olfaction may play a greater role in intimacy and attachment rather than in sexual arousal itself.

Understanding individual differences in olfactory sensitivity could reveal whether certain people rely more on scent in sexual contexts than others. Moreover, given the potential role of smell in intimacy, future studies should differentiate between individuals in long-term relationships and those engaging in casual relationships, as anosmia may affect partner bonding and sexual attraction differently depending on relationship duration and dynamics.

## Figures and Tables

**Figure 1 ijerph-22-01597-f001:**
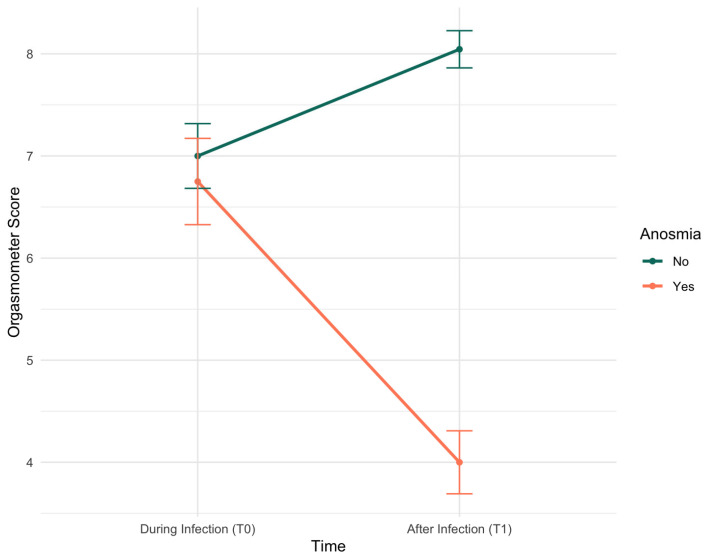
Mean Orgasmometer scores by anosmia condition over time. The graph shows a significant interaction effect, where the group without anosmia demonstrates a recovery and improvement in scores, while the group with anosmia showed a decline from baseline (T0) to follow-up (T1). Error bars represent the standard error of the mean.

**Table 1 ijerph-22-01597-t001:** Demographics characteristics of study subjects with and without anosmia.

Variables	With Olfactory Symptoms (*N =* 61)	Without Olfactory Symptoms (*N* = 77)	χ^2^	*p*
**Gender**			0.34	0.560
*Male*	17 (12.32%)	25 (18.12%)		
*Female*	44 (31.88%)	52 (37.68%)		
**Education**			0.13	0.720
*Without Degree*	24 (17.39%)	28 (20.29%)		
*With Degree*	37 (26.81%)	49 (35.51%)		
** *Relational Status* **			2.45	0.117
*In a relationship*	55 (39.85%)	62 (44.93%)		
*Married*	6 (4.35%)	15 (10.87%)		
** *Occupation* **			0.61	0.436
*Student/Unemployed*	31 (22.46%)	34 (24.64%)		
*Employed*	30 (21.74%)	43 (31.16%)		
*Self-referred symptoms*				
*Anxiety*	45 (32.61%)	49 (35.51%)	1.61	0.205
*Depression*	19 (13.77%)	25 (18.11%)	0.03	0.985

**Table 2 ijerph-22-01597-t002:** Linear mixed-effects model results for Orgasmometer score, showing the effects of time, anosmia, and covariates.

Variable	Estimate	SE	t	df	*p*
Intercept	6.590	0.827	7.967	22.072	<0.001
Time	1.039	0.394	2.641	56.086	0.011
Anosmia	−0.722	0.521	−1.385	18.366	0.183
Gender	0.561	0.369	1.521	44.389	0.135
Age	−0.001	0.025	−0.020	15.566	0.984
Anxiety	−0.170	0.350	−0.486	140.015	0.628
Depression	0.461	0.346	1.332	112.847	0.185
Time*Anosmia	−2.941	0.596	−4.932	51.961	<0.001

## Data Availability

The original contributions presented in this study are included in the article. Further inquiries can be directed to the corresponding authors.

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
