# Peer review of "The Impact of Anosmia on Intimacy: A Linear Mixed-Effects Model Analysis of Sexual Wellbeing Following COVID-19"

_ijerph, 2025, doi:10.3390/ijerph22101597_

Round 1

Reviewer 1 Report

Comments and Suggestions for Authors

Does the sense of smell have a role in intimacy? An observational study evaluating the impact of anosmia on relational wellbeing

The manuscript addresses an interesting and underexplored topic, the role of olfaction in intimacy and sexual functioning, but there are several critical issues that need to be addressed to strengthen its contribution. Most notably, the novelty and theoretical innovation of the study are insufficiently articulated in the introduction, and the relevance of its COVID-19-related focus needs to be justified more clearly given the current context. Additionally, the small sample sizes in several of the subgroups  raise concerns regarding statistical power and the reliability of group comparisons. Addressing these issues would significantly improve the clarity, relevance, and scientific rigor of the manuscript.

1.Abstract-I recommend revisiting the manuscript title, particularly the use of the term "observational study." It would be helpful to clarify why the study is categorized as observational and how that reflects the research design. Additionally, the abstract should clearly specify the exact time points of data collection in relation to the COVID-19 timeline. Given that the pandemic is no longer an acute global crisis, the authors should also address the current relevance of the findings. Moreover, the conclusion presented in the abstract should be more cautiously worded and strictly limited to the findings derived from the study. Overgeneralizations should be avoided.

Introduction

2.The D.E.P.O.R. model, mentioned in the study’s aims, is not reviewed or explained in the introduction. It is recommended to briefly present this model earlier in the manuscript and clarify its relevance to the theoretical framework of the study.

3.The main objective and novelty of the study are only introduced at the end of the introduction. For clarity and coherence, the authors are encouraged to present the study's aim and its innovative contribution earlier in the introduction, even briefly, to better guide the reader.

4.Given the declining public and academic focus on the direct effects of COVID-19, the relevance of the study's context may be questioned. The authors are requested to address this issue, either by justifying the continued significance of COVID-19-related olfactory dysfunction or by highlighting the broader implications of the findings for understanding the role of olfaction in sexual functioning beyond the pandemic context.

Materials and Methods

5.The division of the sample into four subgroups has resulted in at least one group (e.g., males with olfactory symptoms, N=17) with fewer than 30 participants, which raises serious concerns regarding statistical power and the validity of group comparisons. The authors are encouraged to address this limitation and consider its implications for the interpretation of the findings.

6.It is unclear whether the measure used to assess olfactory symptoms was a validated questionnaire or a tool developed ad hoc for this study. The authors should specify the origin of this measure, its structure, and whether it has been validated in previous research.

7.The manuscript lacks critical information on the questionnaires used (e.g., IIEF-15, FSFI, Orgasmometer): Who developed them? What response scales were used? Were sample items included? What is known about their psychometric properties (e.g., reliability, validity)? These details are essential for evaluating the methodological rigor of the study.

  1. In the "Statistical Analysis" section, the authors are requested to clearly specify the name and version of the statistical software used, preferably at the beginning of the section, to ensure transparency and reproducibility.

Results
9.Due to the small sample sizes within the subgroups and the simplicity of the statistical analyses applied, it is unfortunately difficult to fully evaluate the findings presented in the results section. In subgroups where sample sizes permitted meaningful comparisons, more robust statistical methods, such as Pearson correlations and stepwise regression analyses,should have been employed to deepen the understanding of the relationships between variables and enhance the overall rigor of the findings.

Author Response

The manuscript addresses an interesting and underexplored topic, the role of olfaction in intimacy and sexual functioning, but there are several critical issues that need to be addressed to strengthen its contribution. Most notably, the novelty and theoretical innovation of the study are insufficiently articulated in the introduction, and the relevance of its COVID-19-related focus needs to be justified more clearly given the current context.

We would like to thank the Reviewer for this comment. We have now better specified, in the revised manuscript, the importance of studying the role of olfaction, and the COVID-related anosmia in psycho-sexual wellbeing.

Additionally, the small sample sizes in several of the subgroups raise concerns regarding statistical power and the reliability of group comparisons. Addressing these issues would significantly improve the clarity, relevance, and scientific rigor of the manuscript.

We would like to thank the Reviewer for this comment. We understand that selective inclusion criteria (specifically, being sexually active during the COVID pandemic), and the gender division in two groups, had reduced significantly our sample size. However, we believe that in the actual form of the revised manuscript, we have overcome this limit. Following the reviewers’ comments, we have decided to restrict the focus of the manuscript to one aspect of sexual activity, that is, orgasm, and have adopted a stronger and accurate statistical methodology to analyze the data. We hope that the Reviewer will appreciate our effort in making the manuscript more “scientifically sound”.

1.Abstract-I recommend revisiting the manuscript title, particularly the use of the term "observational study." It would be helpful to clarify why the study is categorized as observational and how that reflects the research design.

Thank you very much for your comment. In the revised manuscript, we have changed the title following the new perspective we had given to the study.

Additionally, the abstract should clearly specify the exact time points of data collection in relation to the COVID-19 timeline. Given that the pandemic is no longer an acute global crisis, the authors should also address the current relevance of the findings.

Thank you. The abstract has been corrected following your comment.

Moreover, the conclusion presented in the abstract should be more cautiously worded and strictly limited to the findings derived from the study. Overgeneralizations should be avoided.

We agree with the reviewer’s comment. The conclusion based on the results has been softened

Introduction

2.The D.E.P.O.R. model, mentioned in the study’s aims, is not reviewed or explained in the introduction. It is recommended to briefly present this model earlier in the manuscript and clarify its relevance to the theoretical framework of the study.

We would like to thank the Reviewer for this comment. We have specified the model in the revised manuscript.

3.The main objective and novelty of the study are only introduced at the end of the introduction. For clarity and coherence, the authors are encouraged to present the study's aim and its innovative contribution earlier in the introduction, even briefly, to better guide the reader.

Thank you for your comment. The introduction has been changed following your suggestion.

4.Given the declining public and academic focus on the direct effects of COVID-19, the relevance of the study's context may be questioned. The authors are requested to address this issue, either by justifying the continued significance of COVID-19-related olfactory dysfunction or by highlighting the broader implications of the findings for understanding the role of olfaction in sexual functioning beyond the pandemic context.

We would like to thank the reviewer for the comment. The importance of studying olfaction in relation to sexuality has been highlighted in the text.

Materials and Methods

5.The division of the sample into four subgroups has resulted in at least one group (e.g., males with olfactory symptoms, N=17) with fewer than 30 participants, which raises serious concerns regarding statistical power and the validity of group comparisons. The authors are encouraged to address this limitation and consider its implications for the interpretation of the findings.

We are grateful to the Reviewer for highlighting this critical limitation in our original manuscript. We fully agree that the small subgroup sizes severely compromised the statistical power and the reliability of the group comparisons. To address this fundamental issue, we have completely revised our analytical strategy. We have discarded the subgroup analysis and instead employed a Linear Mixed-Effects Model (LMM). This approach allows us to analyze the entire study sample (N=138) within a single, unified model, thereby maximizing statistical power. Variables such as "anosmia" and "gender" are now included as fixed effects (predictors) rather than being used to split the sample, which provides much more reliable and robust estimates of their influence on the outcome.

6.It is unclear whether the measure used to assess olfactory symptoms was a validated questionnaire or a tool developed ad hoc for this study. The authors should specify the origin of this measure, its structure, and whether it has been validated in previous research.

We thank the Reviewer for this important question. The assessment of olfactory symptoms was based on self-report questions developed ad hoc for this study, which asked participants to report on their ability to detect and recognize their partner's odor. We acknowledge that the lack of an objective, validated olfactory assessment (such as the Sniffin' Sticks test) is a limitation of our study. We have clarified the nature of our measure in the "Materials and Methods" section and have discussed this limitation in the "Limits and future research" section of the manuscript.

7.The manuscript lacks critical information on the questionnaires used (e.g., IIEF-15, FSFI, Orgasmometer): Who developed them? What response scales were used? Were sample items included? What is known about their psychometric properties (e.g., reliability, validity)? These details are essential for evaluating the methodological rigor of the study.

We thank the Reviewer for pointing out this omission. In our revised manuscript, we have shifted the focus of our analysis to the Orgasmometer scores as the primary outcome variable, as this allowed for a more direct and powerful test of our hypothesis. Accordingly, we have substantially expanded the description of this instrument in the "Measures" subsection. We now include details on its developers, its structure as a single-item visual analogue scale, and its validated psychometric properties, including high reliability and its ability to discriminate between functional and dysfunctional populations. To streamline the manuscript and focus on our main findings, the results from the IIEF-15 and FSFI general scores are no longer central to the analysis. Results from other sexological questionnaires will be discussed in future research.

In the "Statistical Analysis" section, the authors are requested to clearly specify the name and version of the statistical software used, preferably at the beginning of the section, to ensure transparency and reproducibility.

We thank the Reviewer for this suggestion. To ensure transparency and reproducibility, we have now specified the statistical software and its version at the beginning of the "Statistical analysis" section. The text now reads: "All statistical analyses were performed using the R software, version 4.5.0 [29]".

Results

9.Due to the small sample sizes within the subgroups and the simplicity of the statistical analyses applied, it is unfortunately difficult to fully evaluate the findings presented in the results section. In subgroups where sample sizes permitted meaningful comparisons, more robust statistical methods, such as Pearson correlations and stepwise regression analyses,should have been employed to deepen the understanding of the relationships between variables and enhance the overall rigor of the findings.

We fully agree with the Reviewer's assessment that our original analyses were too simplistic and lacked the robustness required to draw firm conclusions. Taking this and other comments into account, we have undertaken a complete revision of our analytical approach. We have replaced the previous series of simple tests with a Linear Mixed-Effects Model (LMM). This is a more sophisticated and powerful statistical method that is specifically designed for longitudinal data like ours. The LMM not only addresses the issue of small sample sizes but also allows us to investigate the complex interplay between variables over time, particularly through the inclusion of an interaction term. We are confident that this new approach has substantially enhanced the scientific rigor of our study and provides a much deeper understanding of the data, as recommended by the reviewer.

Reviewer 2 Report

Comments and Suggestions for Authors

Congratulations on the issue of this paper. The text is well-written, but some weaknesses were found in the methods and results. Thus, some improvements are suggested, as detailed in the attached file.

Good work.

Author Response

This paper presents an interesting topic, is appropriate for the journal, and is well organised and well-written. However, some improvements are needed in some sections, as described below.

We would like to thank the Reviewer for Her/his appreciation.

In the whole text, the authors should decide on the use of “Covid-19”, “COVID” or “Covid” because, since the abstract, these different terms are used and, in my opinion, the most correct is COVID-19.

Thank you for this comment. The manuscript has been amended.

The abstract synthesizes the whole text in appropriate detail. Moreover, in addition to the use of different words for COVID-19, it is also suggested to present the type of study in this section.

Thank you for your comment. The abstract has been amended.

The introduction presents the issue in a good way, well understandable, and supported by appropriate literature. This section ends with the study aims, which is correct.

We would like to thank the Reviewer for this positive comment.

The methods section could be improved. The type of study and the momento f data collection should be explicit. On line 97 we can read “A total of 259 participants was recruited, 167 females and 92 males (Figure 1). Nevertheless, the content of Figure 1 is not according to this. The second paragraph refers to the sample was divided ... and the four groups sum a total of 138 participants, which are characterized in table 1. However, it is not clear how and why these 138 were extracted from the 259.

We thank the Reviewer for highlighting these points of confusion. We have substantially revised the "Participants and procedure" subsection to improve clarity. We have removed the original Figure 1, which was misleading. We now explicitly state the inclusion criteria that were used to derive the final analytical sample of 138 participants from the initial 259 recruits. Specifically, the text now clarifies that only participants who were sexually active in the 4 weeks prior to data collection were included in the final analysis, as this was essential for the study's purpose.

Regarding the statistical analysis, some incoherence related to the tests is observed. The authors used the X 2 test to compare differences between groups and, in my opinion, the parametric T test could be applied here. Following, the Mann-Whitney test is referred for unpaired and paired data, and this is a non-parametric test appropriate for continuous variables. However, the two way mixed ANOVA was used ... and this is a parametric test. In addition, I believe that the type of sample and the sampling process do not meet the requirements for applying any ANOVA test. I suggested the Kruskal-Wallis test in this case.

We are very grateful to the reviewer for this astute and detailed critique of our original statistical methods. We fully acknowledge that the combination of different parametric and non-parametric tests was incoherent and that the application of a standard ANOVA to our data was questionable. Based on this crucial feedback, we have completely abandoned the previous analytical strategy. We have now adopted a single, unified, and more appropriate statistical framework by employing a Linear Mixed-Effects Model (LMM). This approach offers several key advantages that directly resolve the issues raised. It provides a single, coherent, and statistically rigorous framework for the entire analysis, eliminating the previous problematic mix of tests. As the state-of-the-art method for repeated-measures data, the LMM correctly accounts for the non-independence of observations from the same participant by including random effects and is more flexible with its statistical assumptions than a traditional ANOVA. Most importantly, this more powerful approach allowed us to test for the interaction between time and anosmia, revealing the core, nuanced finding of our study that was completely missed by the prior analyses. We are confident that this new, more rigorous approach has resolved the statistical incoherence and has substantially strengthened the validity of our findings.

Results after Figure 1 refer to a total sample varying in 256, 257 ... participants, which is different from the initial total sample (259) and from the extracted sample (138). The results presented in table 2 are confusing because the total are different. In addition, any of the p values presented show significant diferences, and instead of p, the other value of the correspondente test should be presented. Table 3 again presents values based on the subsample of 138. So, methods and results need to be improved or better clarified.

Thank you for your comment. In the revised manuscript, all the methods and results sections were corrected following the reviewer’s comments.

The discussion is according to the results presented. Conclusions include limitations in a good way, as well as future research, which is good. So, I hope my suggestions can be useful and have helped to improve the text.

Another time, we are very grateful to the Reviewer for the time spent in revising our manuscript. We hope that in this actual form the manuscript can be considered more suitable for a publication.